# Novel Treatments of Uveal Melanoma Identified with a Synthetic Lethal CRISPR/Cas9 Screen

**DOI:** 10.3390/cancers14133186

**Published:** 2022-06-29

**Authors:** Kseniya Glinkina, Arwin Groenewoud, Amina F. A. S. Teunisse, B. Ewa Snaar-Jagalska, Aart G. Jochemsen

**Affiliations:** 1Department of Cell and Chemical Biology, Leiden University Medical Center, 2300 RC Leiden, The Netherlands; k.a.glinkina@lumc.nl (K.G.); a.f.a.s.teunisse@lumc.nl (A.F.A.S.T.); 2Department of Molecular Cell Biology, Institute of Biology, Leiden University, 2300 RC Leiden, The Netherlands; a.groenewoud@biology.leidenuniv.nl (A.G.); b.e.snaar-jagalska@biology.leidenuniv.nl (B.E.S.-J.)

**Keywords:** uveal melanoma, everolimus, CRISPR-Cas9, DNA-PK, IGF1R

## Abstract

**Simple Summary:**

We performed a CRISPR-Cas9 synthetic lethality screen in order to identify molecular targets whose inhibition would synergistically enhance the effect of everolimus in uveal melanoma cells. *IGF1R* and *PRKDC,* among others, were identified as hits. We verified these hits effects genetically: we treated the uveal melanoma cell lines depleted of *PRKDC* or *IGF1R* with everolimus and, in case of *IGF1R*, observed a synergistic effect. Additionally, we found synergistic growth inhibition with the inhibitors targeting DNA-PKcs or IGF1R in combination with everolimus. Moreover, we investigated the combination of targeted inhibitors of DNA-PKcs and IGF1R with everolimus on uveal melanoma in an in vivo model. The dual DNA-PKcs/mTOR inhibitor CC-115 demonstrated activity in vivo.

**Abstract:**

Currently, no systemic treatment is approved as the standard of care for metastatic uveal melanoma (UM). mTOR has been evaluated as a drug target in UM. However, one of the main limitations is dose reduction due to adverse effects. The combination of everolimus with another targeted agent would allow the reduction of the dose of a single drug, thus widening the therapeutic window. In our study, we aimed to identify a synergistic combination with everolimus in order to develop a novel treatment option for metastatic UM. We exploited CRISPR-Cas9 synthetic lethality screening technology to search for an efficient combination. *IGF1R* and *PRKDC* and several other genes were identified as hits in the screen. We investigated the effect of the combination of everolimus with the inhibitors targeting IGF1R and DNA-PKcs on the survival of UM cell lines. These combinations synergistically slowed down cell growth but did not induce apoptosis in UM cell lines. These combinations were tested on PDX UM in an in vivo model, but we could not detect tumor regression. However, we could find significant activity of the dual DNA-PKcs/mTOR inhibitor CC-115 on PDX UM in the in vivo model.

## 1. Introduction

Uveal melanoma (UM) is the most common intraocular malignant tumor in adults. The incidence of UM is 2–8 cases per million per year in Europe, and approximately 50% of the patients eventually develop metastases [1,2]. The metastases predominantly target the liver. The other sites are the lungs, skin/soft tissue, and bones [3]. Metastatic UM is an aggressive disease: the median survival after the diagnosis is estimated to be 6–12.5 months [4,5].

Currently, no systemic treatment is approved as the standard of care for metastatic UM. The therapeutic options are restricted to local treatment, such as isolated liver perfusion with the chemotherapeutic agent melphalan, which demonstrates a response rate of more than 50% and leads to a significant regression of the lesions. However, this treatment may be applied only to a limited cohort of the patients, and the recurrence of the metastases after the procedure is frequent [6,7,8,9]. Therefore, alternative strategies for the treatment of metastatic UM are being investigated.

The development of novel therapeutic approaches may be guided by the distinct mutational landscape of UM. More than 90% of the tumors harbor activating mutations in genes *GNAQ* and *GNA11,* which encode Gα subunits of trimeric G proteins [10,11]. Activating mutations in the G-protein coupled receptor *CYSLTR2* or in the important Gα signal mediator *PLCB4* have been identified in the rest of UM cases [12,13]. In addition to the deregulated Gα signaling cascade, the UM metastases are frequently characterized by aberrations in the copy numbers of chromosomes 1, 3, 6, and 8 [14]. The gain of 8q and monosomy of chromosome 3 and the subsequent loss of the expression of the BAP1 protein are considered to be the indicators of the higher aggressiveness of the tumor, but the precise molecular mechanisms are not yet clear [10,15,16].

The constant activation of the G-protein signaling cascade leads to the deregulation of multiple signaling routes [17]. The mutant Gαq and Gα11 cause the phosphorylation of PKC, which subsequently activates the MAPK pathway [18]. An alternative role of PKC is the activation of the NF-κB transcriptional program [19,20]. The nuclear translocation and activation of another important transcription factor, YAP1, is triggered by mutant Gαq or Gα11 via the guanine nucleotide exchange factor Trio and its downstream GTPases Rho and Rac [18,21]. The roles of these molecular pathways in the uncontrolled proliferation of UM cells rationalizes the clinical investigation of several targeted drugs, such as MEK inhibitors [22,23,24] and PKC inhibitors [25,26]. Strategies to target YAP1 activity could be of clinical importance as well, but the agents specifically targeting YAP1 are still under development, so no clinical studies have been initiated to date.

Another pathway important for the proliferation of UM cells is the PI3K/AKT/mTOR axis [27,28,29], although it is not directly stimulated by mutant Gαq or Gα11 [30]. Instead, the receptor tyrosine kinases, such as c-MET [31], c-KIT [32], and VEGFR [33], whose signaling is upregulated in UM, have been reported to activate the AKT/mTOR pathway. The loss of PTEN expression also contributes to AKT activation [34].

mTOR has been evaluated as a drug target in UM: the combinations of the rapamycin analog mTORC1 inhibitor everolimus with either PI3K or PKC inhibitors have been shown to inhibit the growth of UM cell lines and reduce tumor volume in PDX models [35,36]. Everolimus is applied in the clinic for various types of cancer [37,38,39].

Active research and clinical trials of everolimus demonstrate its prospect as a potential therapy for metastatic UM. However, one of the main limitations is dose reduction due to adverse effects. The combination of everolimus with another targeted agent would allow the reduction of the dose of a single drug, thus widening the therapeutic window.

An inhibition of IGF1R signaling might overcome the resistance to mTOR inhibition in UM, as suggested by Shoushtari et al. [40]. In their work, the somatostatin analog pasireotide was used to inhibit the activity of somatostatin receptors, which induced the activity of IGF1R by increasing the levels of IGF1 in the plasma. Unfortunately, the clinical benefit of this combination was limited, and the patients experienced a number of adverse effects.

In our study, we aimed to identify a synergistic combination with everolimus in order to develop novel treatment option(s) for metastatic UM. We exploited CRISPR-Cas9 synthetic lethal screening technology to search for an efficient combination.

## 2. Materials and Methods

### 2.1. Cell Culture

OMM2.5, OMM2.3 (a gift of Bruce Ksander), and OMM1 [41] cell lines were cultured in a mixture of RPMI and DMEM-F12 (1:1) supplemented with 10% FBS and antibiotics. MM28, MP38, MP46, and MM66 cells [28] were cultured in IMDM supplemented with 20% FBS and antibiotics. HEK293T cells were cultured in DMEM supplemented with 10% FBS and antibiotics. The cell lines were maintained in a humidified incubator at 37 °C with 5% CO_2_.

### 2.2. Cell Viability Assay

The cells were seeded in triplicate in clear, flat-bottomed 96-well plates in appropriate concentrations (MP46 and MM66: 4000 cells/well, MP38: 6000 cells/well, OMM2.3 and OMM2.5: 2000 cells/well, OMM1: 1500 cells/well). The next day, the medium was supplemented either with a single compound or a combination of the drugs. Five days later, the number of viable cells was determined using a CellTiter-Blue cell viability assay (Promega, Fitchburg, WI, USA). Everolimus, KU-0063794, OSI-906, CC-115, CRT0066101, and KU-57788 (NU7441) were purchased from Selleck Chemicals (Houston, TX, USA), LDN-193189 was obtained from Axon Medchem (Groningen, The Netherlands), and NU7026 was obtained from Cayman Chemical (Ann Arbor, MI, USA).

The synergistic score was calculated according to the Excess over Bliss model [42] using the formula:Excess over Bliss = (Fa_1+2_ − [(Fa_1_ + Fa_2_) − (Fa_1_ × Fa_2_)]) × 100,(1)
where Fa is the fractional activity.

### 2.3. Treatments of Samples for Western Blot

Everolimus: The cells were seeded into 6-well plates (OMM1 and OMM2.3: 3 × 10^5^ cells/well; MM28, MP46, and MM66: 4 × 10^5^ cells/well). The next day, the media were supplemented with 10 nM everolimus, and 24 h later the cells were collected for protein analysis.

IGF1: MP46 cells (4 × 10^5^/well) were seeded into 6-well plates in full serum in the morning and switched to 0.1% serum later in the afternoon. The next day, the cells were pre-treated with OSI-906 (4 µM) or vehicle for 6 h and subsequently stimulated with IGF1 (15 µg/mL; 20 min) then collected for protein analysis.

Bleomycin: MP46 cells (4 × 10^5^ cells/well) were seeded into 6-well plates. The next day, the cells were pre-treated with either 2 µM CC-115, 10 µM NU7026, 5 µM KU57788, or 1 µM KU0063794 for 2 h, then 2 µM bleomycin was added, and after 4 h the cells were harvested for protein analysis.

### 2.4. Western Blot

The cells were rinsed two times with ice-cold PBS and scraped and lysed with Giordano buffer (50 mM Tris-HCl pH = 7.4, 250 mM NaCl, 0.1% Triton X-100, 5 mM EDTA, supplemented with phosphatase and protease inhibitors). Equal protein amounts were separated on SDS-PAGE and blotted on PVDF membranes (Millipore, Darmstadt, Germany). The membranes were blocked with 10% non-fat dry milk in TBST (10 mM Tris-HCl pH = 8.0, 150 mM NaCl, 0.2% Tween-20) and incubated with the primary antibodies diluted in 5% BSA/TBST + phosphatase inhibitors overnight at 4 degrees. The membranes were washed with TBST and incubated with HRP-conjugated secondary antibodies (Jackson Laboratories, Bar Harbor, ME, USA). The chemiluminescent signal was visualized using a Chemidoc machine (Biorad, Hercules, CA, USA).

Primary antibodies: pS473-AKT (#4060), AKT (#2920), pT389-p70S6K (#9205), p70S6K (#2708), pS2056- DNA-PKcs (#68716), IGF1Rβ (#9750) were purchased from Cell Signaling Technology Beverly, MA, USA; vinculin (V9131) was purchased from Sigma-Aldrich, St Louis, MO, USA; DNA-PKcs (sc-390849) and p53 (sc-126) were purchased from Santa Cruz Biotechnology, Dallas, TX, USA; and USP7 (A300-0330A-M) was purchased from Bethyl laboratories, Montgomery, TX, USA.

### 2.5. Flow Cytometry

The cells were seeded into 6-well plates (MM66 and MP46: 1.5 × 10^5^ cells/well, OMM2.5: 1 × 10^5^ cells/well for 5-day treatment; MM66 and MP38: 4 × 10^5^ cells/well for 24 h treatment). The next day, the media were supplemented with a single drug or a combination. After 5 days (24 h for the experiment in Figure 1C), the cells (including floating cells) were collected with trypsinization, washed two times with ice-cold PBS, and fixed with 70% ethanol overnight at 4 °C. The cells were washed with PBS containing 2% FBS, suspended in PBS containing 2% FBS, 50 µg/mL propidium iodide, and 50 µg/mL RNAse A, and incubated for 30 min at 37 °C. Flow cytometry was performed using the BD LSR II system (BD Bioscience, San Diego, CA, USA). In total, 20,000 (10,000 for the experiment in Figure 1C) events were recorded. The data were analyzed using FlowJo v 10.6.1 software. The sum of the G1, S, and G2 percentages was set to 100. The subG1 population was determined as a percentage of the whole population.

### 2.6. CRISPR/Cas9 Synthetic Lethality Screen

MM66 cells (30 million) were infected with a lentiviral sgRNA kinome library [43] at a low multiplicity of infection (MOI = 0.3). The library contained 5971 sgRNAs targeting the full set of human kinases with a coverage of at least 10 sgRNAs per gene. The next day, the media were supplemented with 1 mg/mL puromycin. After 7 days of puromycin selection, the cells were divided into three equal subsets containing approximately 6 million cells (1000× coverage). A “Day 0” subset of cells was cryopreserved in 90% FBS and 10% DMSO for later processing. The media of the treatment subset were supplemented with 10 nM everolimus; an equal amount of DMSO was added to the control subset. The treatment was refreshed every other day, and the coverage was maintained at >1000× for the duration of the experiment. After 10 days of treatment, the cells were scraped and centrifuged at 200 *g*. Genomic DNA was extracted from the resulting cell pellets using a Blood and Cell Culture DNA Kit (Qiagen, Hilden, Germany) according to the manufacturer protocol.

For every sample, a maximum of 20 µg of genomic DNA was divided over twenty 25 µL PCR reactions using barcoded forward primers to be able to deconvolute multiplexed samples after next-generation sequencing. Primer sequences:

Forward primer:

5′-GATGTGTATAAGAGACAGGCTTTATATATCTTGTGGAAAGGACGAAACACC-3′

Reverse primer:

5′-CGTGTGCTCTTCCGATCTCCGACTCGGTGCCACTTTTTCAA-3′

PCR 1 mixture per reaction: 5 µL of Q5 Buffer (NEB, Ipswich, MA, USA), 0.5 µL of 10-mM dNTP, 0.625 µL of 10 µM forward primer 1, 0.625 µL of 10 µM reverse primer 1, 0.25 µL of Q5 High-Fidelity DNA polymerase (NEB), 5 µL of Q5 High GC Enhancer (NEB), 1 µL of SYBR Green (Thermo Fisher, Waltham MA, USA), adding mQ and template to 25 µL. Cycling conditions for PCR 1: 2′ 98 °C, 15 × (15″ 98 °C, 20″ 65 °C, 10″ 72 °C), 14 × (10″ 98 °C, 20″ 72 °C). The products of all reactions were pooled, and 2 µL of this PCR 1 product was diluted 5X with mQ and used in a subsequent PCR 2 reaction using standard Illumina (San Diego, CA, USA) primers containing adapters for next-generation sequencing. PCR 2 mixture per reaction: 4 µL of Q5 Buffer (NEB), 0.5 µL of 10 mM dNTP, 1 µL of 10 µM primer i7, 1 µL of 10 µM primer i5, 0.25 µL of Q5 High-Fidelity DNA polymerase (NEB), 5 µL of the diluted sample from PCR1, adding mQ to 20 µL. Cycling conditions for PCR 2: 2′ 98 °C, 10 × (15″ 98 °C, 20″ 65 °C, 10″ 72 °C). Next, PCR products were purified using standard PCR purification columns. DNA concentrations were measured and, based on this, samples were equimolarly pooled and subjected to Illumina next-generation sequencing. Mapped read counts were subsequently used as the input for the MAGeCK v. 0.5.7 [44] analysis software package. The final hit list was formed based on MAGeCK output on *p* < 0.05, FDR < 50%, removed common essential genes, and the genes enriched in control the subset between day 0 and day 10.

### 2.7. Generation of CRISPR/Cas9 Knockout Cell Lines

The lentivirus to generate stable Cas9-expressing cell lines was produced by transfecting pKLV2-EF1a-Cas9Bsd-W (Addgene #68343) into HEK293T cells together with packaging vectors (psPax2 and pMD2.G). Transduced cells were selected using blasticidin S.

Two sgRNA sequences per gene of interest (the sequences are presented in Appendix A) and a non-targeting sgRNA (Sigma-Aldrich, St Louis, MO, USA) were cloned into a pU6-gRNA-PGK-Puro-2A-BFP vector. Lentiviral particles were produced by a PEI-mediated transfection of HEK293T cells seeded in 15 cm dishes with vector DNA together with the helper plasmids pMDL-RRE, of pRSV-REV, and pMD2.G. The transfected cells were refreshed the next morning, and the virus-containing cell culture supernatants were collected 48 h after transfection and passed through a 0.45 μm filter. The virus titer was quantified by ELISA measuring HIV p24 (ZeptoMetrix Corp., New York, NY, USA). The Cas9-expressing cell lines were transduced with lentiviruses with MOI = 2 in a medium containing 8 µg/mL polybrene. Transduced cells were selected by puromycin selection (1 µg/mL).

### 2.8. Generation of Metastatic UM-Patient-Derived Zebrafish Xenograft Model

Prior to the drug treatment of zebrafish xenografts, the maximum tolerated doses (MTD) of each individual drug and the MTD of the potential secondary drug were determined. To this end, uninjected zebrafish embryos were subjected to a 2-fold serial dilution from 10 µM to 156.25 nM. The highest volume of solvent was used as a vehicle control. Treatments were refreshed every other day for a total treatment length of 6 days. The MTD was determined as the concentration where at least 80% of the treated embryos survived the treatment.

Metastatic uveal melanoma cells were derived from spXmm66-tomato red spheroid cultures, as described by Groenewoud and colleagues [45]. Cells were processed into a single-cell suspension through trituration and were subsequently filtered through a 30 µm cell strainer (Miltenyi Biotec).

Approximately 200–400 cells were implanted into the blood circulation of green fluorescent blood vessel reporter (Tg(Fli:eGFP) [46] zebrafish larvae at 2 days postfertilization (dpf) as described by Groenewoud et al., 2021 [47]. Approximately 16 h postinjection, all incorrectly engrafted zebrafish embryos were removed. All positively engrafted embryos were divided into 24-wells plates (Corning), with six embryos per well. Drug treatment was started 24 h later by the addition of drug-supplemented zebrafish medium to the zebrafish larvae. The treatments were refreshed every other day for a total treatment duration of 6 days. All embryos were washed and anesthetized with 0.04% tricaine (*w*/*v*) (MS-222, Sigma) prior to stereo microscopic imaging using an MZ16FA fluorescence microscope equipped with a DFC420C camera (Leica, Wetzlar, Germany).

All images were subsequently analyzed using Fiji [48] open-source image analysis software using a custom script [49]. Integrated density measures were taken for each group, and all data were normalized to the vehicle control.

## 3. Results

### 3.1. Everolimus Has Cytostatic Effect on UM Cell Lines

First, we examined the effect of everolimus on the survival of uveal melanoma cell lines. The exploited UM cell lines were either derived from UM metastases or did not express the BAP1 protein, an important predisposing factor for the development of metastases (Table 1).

Everolimus reduced the proliferation of the tested cell lines at the low nanomolar concentration range; however, the dose–response curves reached a plateau at approximately 10 nM, and about 50% of the cells (about 30% in the case of MM66) remained viable, irrespective of the increasing concentrations of the inhibitor (Figure 1A). Everolimus blocked mTOR signaling at low nM concentrations, as phosphorylation p70S6 kinase at Thr389, a direct target of mTORC1, was completely abolished upon 24 h treatment (Figure 1B).

We suggest that the observed plateau in the survival curves might be caused by the fast onset of cell cycle arrest. Indeed, treatment with everolimus caused cell cycle arrest in the G1 phase (Figure 1C) and a minor increase in the number of cells in the subG1 phase, suggesting a cytostatic rather than cytotoxic effect on UM cell lines.

**Figure 1 cancers-14-03186-f001:**
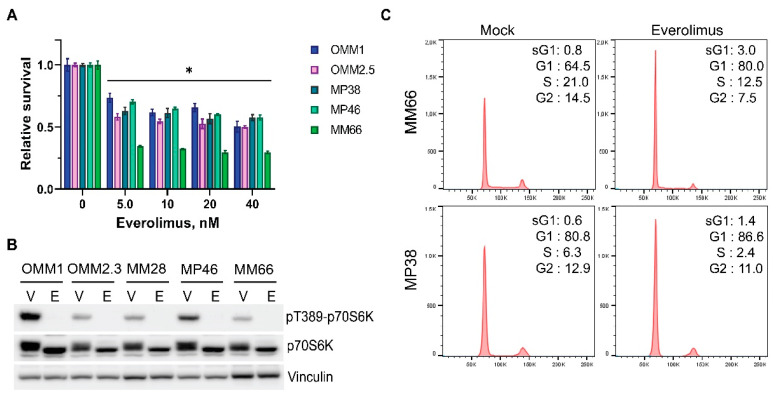
Everolimus inhibits growth of uveal melanoma cell lines by imposing a G1 cell cycle arrest. (**A**) The effect of various concentrations of everolimus on growth of UM cell lines after 5 days of treatment. Significant reduction (*p* < 0.05) of viability compared to a control is indicated with (*). Statistical analysis was performed using a one-way ANOVA with Dunnett’s test for multiple comparisons. Error bars present means ± SEM. (**B**) Treatment with 10 nM everolimus blocks phosphorylation of p70-S6 Kinase at Thr389 after 24 h. The total levels of p70-S6K remain stable. Vinculin represents a loading control. V: vehicle, E: everolimus. (**C**) Treatment with everolimus (10 nM for MM66 and 80 nM for MP38) for 24 h induces cell cycle arrest in G1 phase.

### 3.2. Synthetic Lethal CRISPR-Cas9 Screen

In order to identify the molecular targets whose inhibition would synergistically enhance the effect of everolimus, we performed a synthetic lethal CRISPR-Cas9 screen. We transduced a metastasis-derived MM66 cell line with a library of lentiviruses expressing both the Cas9 gene and sgRNAs targeting the human kinome and subsequently divided the cells into two subsets. One subset was cultured in the presence of everolimus, and the control subset was cultured with an equivalent amount of DMSO. After 10 days of treatment, we compared the pools of sgRNA between these subsets (Figure 2A). In total, 15 genes were significantly under-represented in the everolimus-treated pool compared to the control (the full results of the CRISPR/Cas9 screen are summarized in Appendix A). Using the STRING database, we visualized the interactions between the hits. Several of them are members of the MAPK pathway, and some hits interact with each other directly or via mediators; ACVR1, PRKD3, and DYRK1A do not have any connections to the other hits (Figure 2B).

### 3.3. Inhibitors of DNA-PKcs or IGF1R Synergistically Enhance Cytostatic Effect of Everolimus

Some of the kinases that were selected as hits could readily be targeted by selective inhibitors. However, the interplay between the MAPK family members might form an obstacle for the efficient inhibition of this pathway. We focused on DNA-PKcs (encoded by *PRKDC* gene), IGF1R, ALK-2 (encoded by *ACVR1* gene), and PRKD3, as the targeted inhibitors to these kinases are commercially available and actively investigated in the clinic.

We investigated whether the selected inhibitors could synergistically enhance the effect of everolimus on UM cell survival. The drug combinations were tested on a panel of four UM cell lines, MM66, OMM2.3, MP38, MP46, which all harbor *GNAQ/11* mutations but represent different BAP1 statuses and origins. All the tested combinations reduced the cell viability more than any of the single treatments (Figure 3A–E and Appendix A). We used the Excess over Bliss (EoB) model to determine if the tested drugs acted synergistically (positive EoB score), additively (zero score), or antagonistically (negative EoB score) in combination with everolimus. All the tested combinations demonstrated positive EoB scores across most of the concentrations; in general, the EoB scores of the combinations of everolimus with ALK-2 inhibitor (Figure 3A) or PRKD3 inhibitor (Figure 3B) were lower than the EoB scores of the combinations of everolimus with the DNA-PKcs inhibitors NU7026 (Figure 3D) and KU57788 (Figure 3E) or the IGF1R inhibitor OSI-906 (Figure 3C). We chose the combinations of everolimus with DNA-PKcs inhibitors and IGF1R inhibitor for further investigation.

DNA-PKcs and mTOR belong to the same family of kinases and share a structurally similar catalytic site that allows for the development of an inhibitor targeting both kinases. One such double action inhibitor, CC-115, is currently undergoing clinical trials for patients with advanced solid tumors and hematologic malignancies [51]. We evaluated the sensitivity of the UM cell lines to CC-115. The growth of all the tested cell lines was inhibited after 3 days of treatment with CC-115 in nM concentrations (Figure 3F).

In order to evaluate the target engagement of the selected inhibitors, we probed the downstream targets by Western blotting. As illustrated in Figure 4A, treatment with OSI-906 completely blocked the IGF1-induced phosphorylation of AKT at Ser473 and even reduced basal Ser473-AKT, suggesting efficient inhibition of IGF1R.

To demonstrate the effect of the DNA-PKcs inhibitors, we pre-treated MP46 cells with either CC-115, NU7026, or KU57788 for 2 h, then incubated for 4h with bleomycin, an agent that causes DNA double-strand breaks and activates DNA-PK (Figure 4B). In non-treated MP46 cells, DNA-PKcs is highly expressed but not activated, as indicated by the low signal of phosphorylation at Ser2056. After the addition of bleomycin, the phosphorylation at Ser2056 strongly increased, and the total levels of DNA-PKcs dropped. In the cells pre-treated with either CC-115, NU7026, or KU57788, the level of pS2056-DNA-PKcs after the addition of bleomycin was clearly decreased less compared to in non-pre-treated, bleomycin-treated sample. Treatment with KU57788 could prevent the drop in the total DNA-PKcs level.

Additionally, we probed the pT389-p70S6K and total p70S6K levels in order to clarify the effects of the inhibitors on mTOR signaling (Figure 4B). Incubation with CC-115 completely abrogated the phosphorylation of p70S6K on Thr389, while the other inhibitors reduced the pT389-p70S6K signal to lesser extent. Thus, all three DNA-PKcs inhibitors block DNA-PKcs activation and affect the mTOR signaling route to varying extents.

### 3.4. The Combinations of Everolimus with DNA-PKcs or IGF1R Inhibitors Induce Cell Cycle Arrest in UM Cell Lines

We analyzed the effect of everolimus and DNA-PKcs or IGF1R inhibitors and their combinations on the cell cycle progression of the MM66 (Figure 5), MP46, and OMM2.5 cell lines (Appendix A).

Treatment with the DNA-PKcs inhibitor NU7026 did not change the cell cycle profile in any cell line nor did the IGF1R inhibitor OSI-906. KU57788 showed no effect in MP46 and MM66 cells but induced some accumulation of cells in the G1 phase in OMM2.5 cells. All cell lines treated with the combinations of everolimus with either NU7026, KU57788, or OSI-906 were arrested in G1 without a significant increase in the subG1 fraction. These results indicated that the growth inhibition was caused by cell cycle arrest rather than apoptosis. Similarly, treatment with CC-115 resulted in cell cycle arrest in the G1 phase in MM66 and OMM2.5 cells, but the effect was less pronounced in the MP46 cell line.

### 3.5. mTOR Active Site Inhibitor KU0063794 Is, Similar to Everolimus, Synergistic in Combination with DNA-PKcs or IGF1R Inhibitors

We hypothesized that blocking the mTOR active site (thus blocking the activity of both the mTORC1 and mTORC2 complexes, while everolimus only blocks mTORC1 activity) may further increase the synergistic effect of the combinations with DNA-PKcs or IGF1R inhibitors on cell viability. We evaluated the effect of DNA-PKcs inhibitors NU7026 and KU57788 and the IGF1R inhibitor OSI-906 in combination with active-site mTOR inhibitor KU0063794. The EoB values of the tested DNA-PKcs inhibitors with KU0063794 were comparable or even somewhat less than the ones with everolimus (Appendix A); the synergistic effect of the combination of OSI-906 with KU0063794 was similar to the one with everolimus, as represented in Appendix A.

The target engagement of KU0063794 is shown in Appendix A. Treatment with KU0063794 decreased the phosphorylation of p70S6K at Thr389 and therefore blocked the activation of the downstream target of mTORC1. In the report by Garcia-Martinez et al. (2009), it was demonstrated that KU0063794 blocks the activity of both mTORC1 and mTORC2 [52].

### 3.6. Genetic Verification of the Hits

We also investigated whether a genetic ablation of *PRKDC* or *IGF1R* synergized with everolimus in UM growth inhibition. Therefore, we established derivatives of the OMM2.5, MM66, and OMM1 cell lines to express Cas9 and subsequently introduced lentiviruses expressing sgRNAs targeting either *PRKDC* or *IGF1R* or a non-targeting sgRNA. A significant depletion of DNA-PKcs protein could be reached with sgPRKDC#2 in the MM66 cell line and to a lesser extent in OMM2.5, while the expression of sgPRKDC#1 had no effect (Figure 6A). In OMM1 cells, the expression of combined sgPRKDC#1 and 2 or only sgPRKDC #2 led to substantial drops in DNA-PKcs protein (Figure 6B).

We noticed a significant decrease in the growth rate after the depletion of DNA-PKcs, especially in MM66 and OMM1 cells expressing sgPRKDC#2, indicating the importance of this protein for UM cell proliferation (Figure 6D).

Unfortunately, treating the *PRKDC*-depleted cell lines with everolimus did not affect the sensitivity compared to control cells expressing the non-targeting control (Figure 6F–H). At the same time, we observed that the depletion of DNA-PKcs slightly sensitized the cells for bleomycin (Appendix A), confirming previous reports that the inhibition of DNA-PKcs results in sensitization to DNA-damaging agents [53,54].

A CRISPR/Cas9-mediated knockout of *IGF1R* resulted in an efficient depletion of IGF1R protein levels in all the tested cell lines, as illustrated in Figure 6C. Upon knockout of *IGF1R*, the proliferation rates of OMM2.5 and MM66 cells, but not OMM1 cells, slowed down (Figure 6E). IGF1R depletion made OMM2.5 and OMM1 cells significantly more sensitive to everolimus than the control cells (Figure 6I,K), but we did not observe the same effect in MM66 cells after 6 days of treatment (Figure 6J). However, we could find minor differences in the sensitivity of MM66 after a 4-day treatment. (Appendix A).

According to our preliminary data, the depletion of IGF1R sensitizes UM cells to the active-site mTOR inhibitor KU0063794 (Appendix A).

### 3.7. CC-115 Efficiently Inhibits Growth of UM Cells in an In Vivo Model

Next, we examined whether the synergistic effects of the combinations of everolimus with KU57788 or OSI-906 observed in vitro could be reproduced in vivo. For this purpose, we exploited a metastatic UM-patient-derived zebrafish xenograft model (UM zf-PDX) [45]. This model is based on 3D spheroids grown from Xmm66 metastatic UM PDX cells derived from the same original donor as the MM66cell line. Prior to the treatment, the maximal tolerated dose (MTD) for each drug was determined by treatment of uninjected zebrafish larvae with varying concentrations of each drug by changing the water lace with drugs every other day. Unfortunately, the MTD of KU57788 and OSI-906 turned out to be lower than the concentrations we used for the growth inhibition of UM cells in vitro (Figure 7A).

Even so, we applied the inhibitors, as single agents or in combination with everolimus, to UM zf-PDX and measured the change in tumor burden after 6 days of treatment (Figure 7B). The tumor burden after treatment with either everolimus or OSI-906 was comparable to the control but was slightly increased after the treatment with KU57788. Similarly, the combinations everolimus–KU57788 and everolimus–OSI-906 did not reduce the tumor burden compared to control, which could be explained by the low concentrations of compounds used due to the toxicity.

Interestingly, the double mTOR/DNA-PKcs inhibitor CC-115 was well-tolerated by the zebrafish larvae, and it significantly decreased the tumor burden in the UM zf-PDX.

## 4. Discussion

The inhibition of the PI3K/AKT/mTOR pathway is considered a putative therapeutic strategy for the patients with metastatic uveal melanoma, but monotreatment with the inhibitors targeting specific members of this pathway does not induce apoptosis in UM cells, possibly due to the activation of feedback loops [35,55]. Combining several inhibitors targeting up- and downstream members of the pathway might overcome this activation. For example, the combination of the PI3K inhibitor GDC0941 and the mTOR inhibitor everolimus was demonstrated to suppress tumor growth in a patient-derived xenograft UM model [35].

Compensatory mechanisms, other than the activation of feedback loops, might be linked to the resistance of UM cells to PI3K/AKT/mTOR inhibitors. Since PI3K/AKT/mTOR is only one of numerous pathways activated in UM, the simultaneous blocking of several signaling routes, for instance MAPK and AKT [55], might be essential for apoptosis induction and the control of tumor growth.

In our work, we exploited synthetic lethal CRISPR screening to search for targets that enhance the cytostatic effect of the mTOR inhibitor everolimus. Several members of the MAPK cascade, *MAPK1*, *MAP2K2*, *RAF1*, *BRAF,* and *KSR1*, were present in a resulting hit list. The activation of the MAPK cascade upon everolimus treatment and the synergism of the dual inhibition of these pathways was previously reported in various types of cells [56,57,58]. We did not choose the inhibitors of MAPK signaling for further investigation, as several MEK inhibitors have undergone clinical trials for metastatic UM but were proven inefficient for tumor control [59].

Another hit from our screen, the *IGF1R* gene, is highly expressed in hepatic UM metastases and has been suggested as a putative therapeutic target [60]. The clinical study of a monoclonal antibody against IGF1R, IMC-A12, for the treatment of metastatic UM demonstrated that the agent was well-tolerated and resulted in stable disease in 50% of the patients, but neither a complete nor a partial response was reached in any patient [61].

We investigated the effect of the combination of the IGF1R inhibitor OSI-906 and everolimus on UM cell lines. This combination synergistically slows down cell growth but does not induce apoptosis in UM cell lines. This combination was also tested on the zf-PDX UM model, but we could not detect tumor regression. Due to toxicity of the inhibitors for zebrafish embryos, we had to limit the applied concentrations to levels lower than those needed for the growth inhibition of the UM cell lines in vitro.

The combination of everolimus and agents targeting IGF1R has already been investigated in the clinic for the treatment of advanced solid tumors. The combination of everolimus and figitumumab was well-tolerated and resulted in one partial response in a patient with a malignant solitary fibrous tumor and 15 patients with stable disease out of 18 patients evaluable for response in phase I trials [62]. The phase II trials of the combination of everolimus and OSI-906 for the patients with refractory metastatic colorectal cancer did not indicate a clinical benefit [63].

The combination of everolimus and specific IGF1R inhibitors has not yet been investigated in the clinic for the treatment of metastatic UM. The phase II trials of everolimus combined with the somatostatin analogue pasireotide, which indirectly inhibits IGF1R signaling by affecting IGF1 levels besides affecting a number of other targets, demonstrated a limited clinical benefit. During this study, all patients experienced adverse effects, and 50% of the patients had at least one dose reduction of everolimus [40]. The detected decrease in IGF1 plasma levels caused by pasireotide substantially varied between patients (7–75%). This reduction was exploited as the only readout for IGF1R inhibition, and no correlation of the IGF1 level with the clinical outcome was found, but this might not directly correlate with IGF1R activity.

Like IGF1R, DNA-PKcs has been reported as a potential therapeutic target for UM. Its inhibition has been demonstrated to slow the proliferation of UM cells and sensitize the cells to radio- and chemotherapy [64,65]. A high expression of *PRKDC* is associated with a worse prognosis [66].

Our results indicate that DNA-PKcs inhibitors slow down the growth of UM cells, although rather high concentrations are needed. The tested inhibitors NU7026 and KU57788 caused G1 cell cycle arrest but did not induce apoptosis in the tested cell lines as single treatments or in combination with everolimus. The synergistic effect of the combinations was most likely caused by the faster onset of cell cycle arrest compared to the single treatments.

To verify the results of the CRISPR screen, we compared the sensitivity to everolimus of MM66, OMM1, and OMM2.5 derivatives containing *IGF1R*- or *PRKDC*-knockout (*IGF1R-KO* and *PRKDC-KO*) or non-targeting sgRNA as a control. The survival of *IGF1R-KO* OMM2.5 and OMM1, but not MM66, was significantly reduced by everolimus after a 6-day treatment. The observed difference between the cell lines might be explained by the variance in the culture conditions: OMM2.5 and OMM1 were cultured in medium with the addition of 10% FBS; in the medium for MM66, the concentration of FBS was 20%. The bovine IGF1 present in FBS is identical to human IGF1 and can stimulate IGF1R and insulin receptor, impairing the effect of knockout [67]. Additionally, the duration of the treatment, only half the period of the initial CRISPR screen, might also affect the result because MM66 cells have a longer doubling time than OMM2.5 or OMM1 cells.

We did not observe the difference in sensitivity to everolimus between *PRKDC-KO* cells and OMM2.5, OMM1, and MM66 cells expressing non-targeting sgRNA, despite a previously demonstrated synergistic effect of a DNA-PKcs and mTOR inhibitor combination. In contrast to the inhibitors applied simultaneously, the CRISPR/Cas9 knockouts of *PRKDC* were generated first, after which the cells were treated with everolimus. We noted that *PRKDC-KO* cells, especially MM66, changed morphology and their growth slowed down significantly, possibly due to cell cycle arrest. A knockout of *PRKDC* may cause a more profound effect on the cells than the inhibitors since DNA-PKcs protein, but not its catalytical activity, are essential for various biological functions, such as DNA end processing [68]. Likely, the cells in the cell cycle arrest state could not react to the new stimuli of everolimus. Therefore, we could not detect any additional sensitivity to the inhibitor.

Besides the combinations of DNA-PKcs inhibitors with everolimus, we evaluated the effect of the double DNA-PKcs/mTOR inhibitor CC-115 on UM cell lines. CC-115 binds to the catalytic site of mTOR, thus inhibiting both mTORC1 and mTORC2, while everolimus, as a rapamycin analogue, selectively blocks mTORC1. However, the application of mTORC1/2 inhibitor KU0063794 instead of everolimus does not increase the synergistic effect of the combination, and therefore, this cannot explain the sensitivity of UM cells to CC-115. According to the previous report [69], CC-115 indirectly inhibits ATM activation. Interestingly, ATM-deficient cells are significantly more sensitive than ATM-proficient cell lines, while the effect of CC-115 on ATM-proficient cell lines is synergistically enhanced by DNA-damaging agents.

CC-115 in nM concentrations inhibits the growth of UM cell lines; the mechanism of this inhibition, similar to the aforementioned combination treatment, was primarily cell cycle arrest in the tested conditions. It is possible that a combination with ATM inhibition or DNA damaging agent could induce an apoptotic response. Remarkably, in contrast to the DNA-PKcs inhibitor KU-57788, CC-115 treatment is well-tolerated by zebrafish embryos, and CC-115 very efficiently reduces the tumor burden in the zf-PDX UM model.

The results of the phase I clinical trials of CC-115 in patients with advanced solid and hematologic malignancies looked promising since CC-115 was well-tolerated, and stable disease was reached in 53%, 22%, 21%, and 64% of patients with head and neck squamous cell carcinoma, Ewing sarcoma, glioblastoma multiforme, and castration-resistant prostate cancer, respectively [51]. The phase I trial of CC-115 with enzalutamide demonstrated this combination is safe and active in patients with metastatic castration-resistant prostate cancer [70].

## 5. Conclusions

In summary, we can conclude that the targeted inhibitors of DNA-PKcs or IGF1R synergistically enhance the cytostatic effect of everolimus on UM cell lines. The dual DNA-PKcs/mTOR inhibitor CC-115 efficiently inhibits UM growth in cell culture and in an in vivo model, so it might be suitable for the treatment of metastatic UM patients.

## Figures and Tables

**Figure 2 cancers-14-03186-f002:**
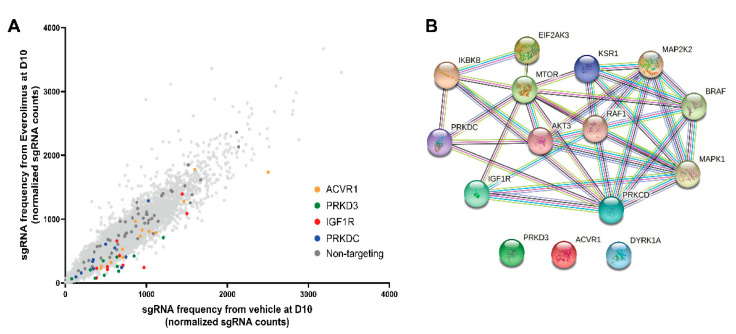
A CRISPR/Cas9 screen indicates *IGF1R* and *PRKDC* as putative synthetic lethal pairs with mTOR inhibition. (**A**) Enrichment of the selected sgRNAs on day 10 versus day 0. The cells were transduced with lentiviral particles expressing both Cas9 and sgRNAs targeting the human kinome and were subsequently treated with 10 nM everolimus or vehicle for 10 days. The frequency of sgRNA representation in everolimus vs. vehicle subsets on day 10 is plotted. Each dot corresponds to a particular sgRNA: yellow dots represent sgRNAs targeting *ACVR1*, green dots represent sgRNAs targeting *PRKD3*, red dots represent sgRNAs targeting *IGF1R*, blue dots represent sgRNAs targeting *PRKDC*, dark gray dots represent non-targeting sgRNAs. (**B**) STRING network of CRISPR/Cas9 screen top hits. Each sphere refers to a protein corresponding to a hit gene. Edges represent associations between the proteins based on known interactions from curated databases (blue), experimentally determined interactions (magenta), or interactions predicted by gene neighborhood (green), gene fusions (red), or gene co-occurrence (dark blue). Other types of interactions are identified by text mining (light green), co-expression (black), and homology (violet).

**Figure 3 cancers-14-03186-f003:**
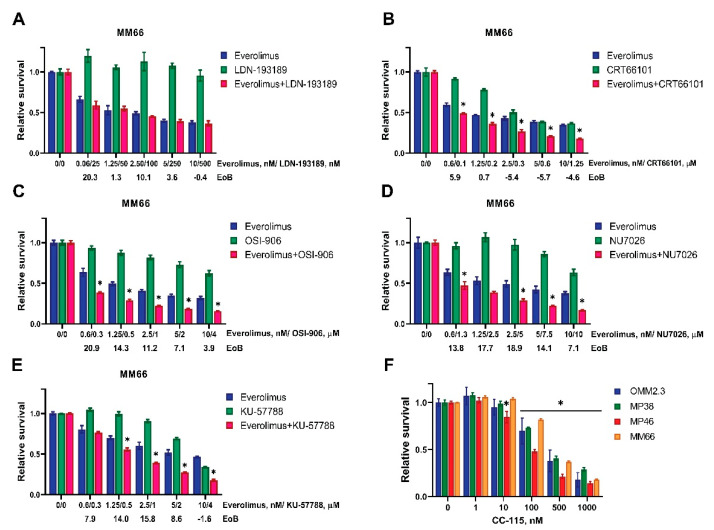
Targeted inhibitors of DNA-PKcs or IGF1R demonstrate synergistic effect in combination with everolimus. (**A**–**E**) MM66 cells were treated with either everolimus (blue bars) or a targeted inhibitor of (**A**) ALK-2 (LDN-193189), (**B**) PRKD3 (CRT66101), (**C**) IGF1R (OSI-906), (**D**) DNA-PKcs (NU7026), (**E**) DNA-PKcs (KU57788) (green bars), or the combination of both (red) for 5 days to determine the effect on viability. The synergistic scores were calculated by the Excess over Bliss algorithm; they are indicated for each pair of concentrations below the X-axes (EoB). Significant reductions (*p* < 0.05) in viability in combinational treatment comparing to both of the single treatments are indicated with (*). Statistical analysis was performed using one-way ANOVA with Tukey’s test for multiple comparisons, and error bars present means ± SEM. (**F**) The effect of various concentrations of double DNA-PK/mTOR inhibitor CC-115 on viability of UM cell lines after 3 days of treatment. Significant reduction (*p* < 0.05) of viability comparing to a control is indicated with (*). Statistical analysis was performed using one-way ANOVA with Dunnett’s test for multiple comparisons, and error bars present means ± SEM. Every experiment was performed at least in three independent biological replicates; a representative replicate is presented.

**Figure 4 cancers-14-03186-f004:**
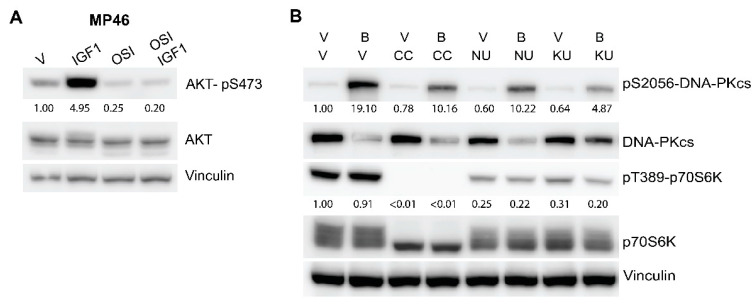
Target engagement of the selected inhibitors. (**A**) Treatment with OSI-906 abolished the IGF1-induced phosphorylation of AKT at Ser473. MP46 cells were pre-treated with 4 µM OSI-906 (OSI) or vehicle (V) for 6 h and subsequently stimulated with 15 ng/mL IGF1 for 20 min. (**B**) Treatment with DNA-PK inhibitors reduced the activation of DNA-PKcs by bleomycin. MP46 cells were pre-treated with either vehicle (V), 2 µM CC-115 (CC), 10 µM NU7026 (NU), or 5 µM KU57788 (KU) for 2 h. After addition of 2 µM bleomycin (**B**) the cells were incubated for 4 h and harvested for analysis. Quantifications were performed with the Image Lab 6.0 program. Raw values were normalized to vinculin values, and subsequently, relative values compared to untreated cells were calculated.

**Figure 5 cancers-14-03186-f005:**
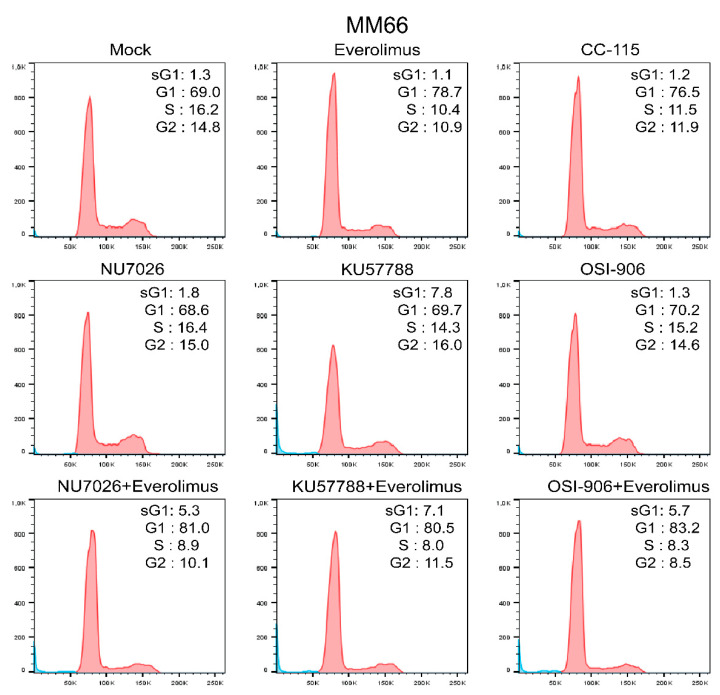
Treatment with everolimus (10 nM) alone or in combination with NU7036 (10 µM), KU57788 (2 µM), or OSI-906 (3 µM) for 5 days induced cell cycle arrest in G1 phase of MM66 cell line. Monotreatment with NU7026, KU57788, or OSI-906 did not affect cell cycle profile. Treatment with CC-115 (300 nM) induced G1 arrest. The number of cells in subG1 (sG1) subset marginally increased upon the treatments.

**Figure 6 cancers-14-03186-f006:**
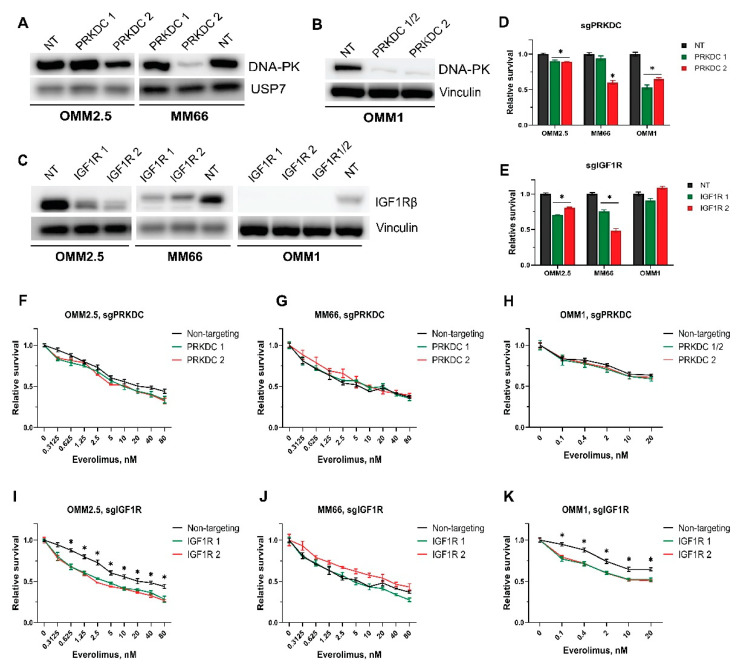
Effect of everolimus in combination with genetic depletion of either *PRKDC* or *IGF1R* on survival of UM cell lines. (A-B) Reduction of DNA-PKcs protein level upon genetic depletion of *PRKDC* by sgPRKDC 2 in OMM2.5 and MM66 cell lines (**A**) or OMM1 (**B**). NT: Non-targeting sgRNA, PRKDC 1, 2: sgRNAs targeting *PRKDC*. (**C**) Reduction of IGF1R protein level upon genetic depletion of *IGF1R* in OMM2.5, MM66, and OMM1 cell lines. NT: Non-targeting sgRNA, IGF1R 1, 2: sgRNAs targeting *IGF1R*. (**D**,**E**) Effect of *PRKDC* knockout (**D**) or *IGF1R* knockout (**E**) on viability of the tested cell lines. The duration of the experiments with OMM2.5 and MM66 was 6 days, and the experiment with OMM1 lasted 5 days. Significant reduction (*p* < 0.05) in viability compared to a control is indicated with (*). Statistical analysis was performed using one-way ANOVA with Dunnett’s test for multiple comparisons. Error bars present means ± SEM. (**F**–**K**) Effect of combination of everolimus treatment with CRISPR/Cas9-mediated knockout of either *PRKDC* (**F**–**H**) or *IGF1R* (**I**–**K**) on viability of UM cells. The viability of OMM2.5 and MM66 cells was assessed after 6 days of treatment, and the viability of OMM1 cells was assessed after 5 days. Asterisk (*) indicates significant differences (*p* < 0.05) between non-targeting control and both sgRNAs. Statistical analysis was performed using one-way ANOVA with Tukey’s test for multiple comparisons. Error bars present means ± SEM. Every experiment was performed in at least three independent biological replicates; a representative replicate is presented.

**Figure 7 cancers-14-03186-f007:**
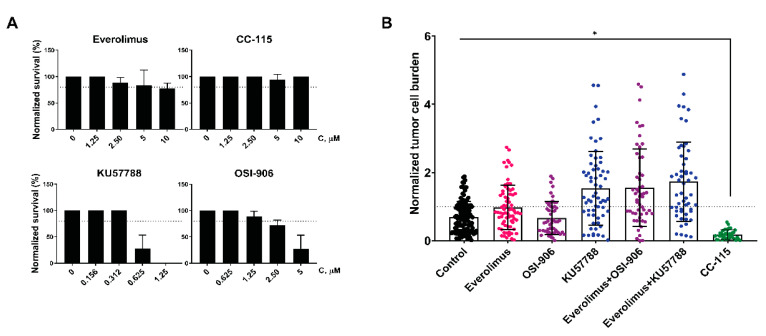
Effect of the DNA-PKcs and IGF1R inhibitors in combination with everolimus on UM-patient-derived zebrafish xenograft model. (**A**) Determination of MTD of everolimus (top left), CC-115 (top right), KU57788 (bottom left), or OSI-906 (bottom right). The zebrafish embryos were subjected to a serial dilution of the inhibitor. Treatments were refreshed every other day for a total treatment duration of 6 days. The MTD was determined as the concentration where at least 80% of the treated embryos survived the treatment. (**B**) Effect of everolimus (1.25 µM), OSI-906 (1.25 µM), KU-57788 (0.3125 µM), and CC-115 (10 µM) alone or combined on tumor burden in zf-PDX UM model. Asterisk (*) indicates significant reduction (*p* < 0.05) compared to a control. Statistical analysis was performed using one-way ANOVA with Dunnett’s test for multiple comparisons.

**Table 1 cancers-14-03186-t001:** Uveal melanoma cell lines.

Cell Line	Origin	GNAQ Mutation	GNA11 Mutation	BAP1 Expression	Reference
OMM2.3	Liver metastases	c.626 A > C	-	Yes	[41,50]
OMM2.5	Liver metastases	c.626 A > C	-	Yes	[28,41,50]
OMM1	Subcutaneous metastasis	-	c.626 A > T	Yes	[28,41,50]
MM66	PDX established from liver metastasis	-	c.626 A > T	Yes	[28]
MM28	PDX established from liver metastasis	-	c.626 A > T	No	[28]
MP46	PDX established fromPrimary tumor	c.626 A > T	-	No	[28]
MP38	Primary tumor	c.626 A > T	-	No	[28]

## Data Availability

The data presented in this study are available in this article (and Appendix A).

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
