# Peer review of "Novel Treatments of Uveal Melanoma Identified with a Synthetic Lethal CRISPR/Cas9 Screen"

_cancers, 2022, doi:10.3390/cancers14133186_

Round 1

Reviewer 1 Report

The authors have presented a paper about "Novel Treatments of Uveal Melanoma Identified with a Synthetic-Lethal CRISPR/Cas9 Screen".

The topic is absoluterly interesting and the authors have demostrated a deep knowledge of the argument.

The introduction is adequate and material and methods section is clearly presented. 

However I beleive that there are some inconsistencies which need to be addressed as follows: in fact the authors state that "These combinations... do not induce apoptosis in UM cell lines" and in another point "These combinations were tested...in vivo model, but we could not detect tumor regression".

However they conclude that " it might be suitable for treatment of metastatic UM patients".

I believe that the conslusion is not supported by these premises either the conlusion should be modified or the reserchers should argue more in detail about it. 

Author Response

Dear reviewer, 

Please see our response in the attachment.

Reviewer 2 Report

            The study by Glinkina et al is an interesting study that focused on exploring new therapeutic targets for uveal melanoma. The authors used different cell lines and made an interesting strategy that consisted of using a lethal CRISPR strategy for a broader evaluation. Several targets were identified, which were then evaluated in follow up in vitro and in vivo experiments. Although the initial findings show potential, the in vivo model data showed disappointing results. In my view, this arises because of several different cell lines and conditions that were used. Nonetheless, the study provides important data for the literature.

The material and methods part does not contain valuable information that in this current stage prevents reproducibility. Surprisingly, the authors do not mention the n number of the experiments. The authors use student t test for the analyzes, which is not appropriate considering the experiment design. Moreover, the figures do not contain any indication of the differences and no quantification data is provided for western blot and FACS data. Without this important information, it is an exceedingly difficult task to follow the manuscript.

In the results part the authors must justify why they choose different cells lines. The authors must explicitly explain why cell lines were used.

Considering these limitations, at this current stage I cannot endorse its publication as there are issues that must be addressed.

Specific points:

M&M

2.2.

Add the cell density for each experiment as this information is crucial for reproducibility.

2.3.

Could the author explain why MP46 cells were seeded in serum?

2.5.

Could the authors explain why FACS experiments were made 5 days later instead of 24 h as previously mentioned?

2.6.

In topic 2.6 could the authors explain why using an FDR < 50%?

Results

I request the authors to provide the evidence or reference of the cell line information provided in table 1.

Figure 1

            Considering the experiment design that consists of different drug concentration, it is not suitable to use Student t test. The authors must use Anova One-Way in Figure 1A and specify in the figure the differences.

            Quantitative analysis of the WB and FACS results must be provided in Figure 1B and C.

            Add the n number for each assay in figure 1.

Figure 2

            Explain why MM66 was chosen for follow up analyses.

            Provide n number.

            Provide in the M&M a description of how Excess over Bliss analyzes were made.

Figure 3

            The authors must use Anova One-Way and indicate the differences in each graph

            Provide n number for each assay

Figure 4

            Western blot analyzes must be followed by quantification

            It is not clear why MP46 cell was chosen for this assay. Please provide this information.

Figure 5

            Provide FACS quantitative analyzes

            Provide n number

            Provide the rationality of why using the selected cell lines

Figure 6 and 7

            Provide n number

Author Response

(The authors gave the same response as above.)

Round 2

Reviewer 1 Report

I have no further comments.